# Palliative Surgery or Metallic Stent Positioning for Advanced Gastric Cancer: Differences in QOL

**DOI:** 10.3390/medicina57050428

**Published:** 2021-04-28

**Authors:** Enrico Fiori, Daniele Crocetti, Paolo Sapienza, Roberto Cirocchi, Antonio V. Sterpetti, Michelangelo Miccini, Marcello Accordino, Silvano Costi, Pierfrancesco Lapolla, Andrea Mingoli, Giorgio De Toma, Antonietta Lamazza

**Affiliations:** 1Department of Surgery “Pietro Valdoni”, “Sapienza” University of Rome, 00161 Rome, Italy; enrico.fiori@uniroma1.it (E.F.); paolo.sapienza@uniroma1.it (P.S.); antonio.sterpetti@uniroma1.it (A.V.S.); michelangelo.miccini@uniroma1.it (M.M.); marcello.accordino@uniroma1.it (M.A.); pierfrancesco.lapolla@uniroma1.it (P.L.); andrea.mingoli@uniroma.it (A.M.); giorgio.detoma@uniroma1.it (G.D.T.); antonietta.lamazza@uniroma1.it (A.L.); 2Department of Surgical and Biomedical Sciences, Perugia University, 06123 Terni, Italy; roberto.cirocchi@unipg.it; 3Department of Economics and Finance, LUISS Guido Carli, 00198 Rome, Italy; silvano.costi@studenti.luiss.it

**Keywords:** gastric cancer, stomach-partitioning gastrojejunostomy, stent positioning

## Abstract

Background and Objectives: Twenty percent of the patients affected with stage IV antropyloric stomach cancer are hospitalized with a gastric outlet obstruction syndrome (GOOS) requiring its resolution to improve the quality of life (QoL). We present our preliminary short- and mid-term results regarding the influence of endoscopic placement of self-expandable metal stent (SEMS) or open stomach-partitioning gastrojejunostomy in QoL. Materials and Methods: In this prospective randomized longitudinal cohort trial, we randomly assigned 27 patients affected with stage IV antropyloric stomach cancer into two groups: Group 1 (13 patients) who underwent SEMS positioning and Group 2 (14 patients) in whom open stomach-partitioning gastrojejunostomy was performed. The Karnofsky performance scale and QoL assessment using the EQ-5D-5L™ questionnaire was administered before treatment and thereafter at 1, 3, and 6 months. Results: At 1-month, index values showed a statistically significant deterioration of the QoL in patients of Group 2 when compared to those of Group 1 (*p* = 0.004; CI: 0.04 to 0.21). No differences among the groups were recorded at 3-month; whereas, at 6-month, the index values showed a statistically significant deterioration of the QoL in patients of Group 1 (*p* = 0.009; CI: −0.25 to −0.043). Conclusions: Early QoL of patients affected with stage IV antropyloric cancer and symptoms of GOOS is significantly better in patients treated with SEMS positioning but at 6-month the QoL significantly decrease in this group of patients. We explained the reasons of this fluctuation with the higher risk of re-hospital admission in the SEMS group.

## 1. Introduction

More than 20% of patients with gastric cancer have at presentation a stage IV disease [1]. Advanced adenocarcinoma of the antro-pyloric region often determines a condition of gastric outlet obstruction syndrome (GOOS), which requires a rapid resolution for the severe consequences that will occur if the obstruction is not resolved. GOOS causes malnutrition, fluid, and electrolyte imbalances that are difficult to control [2]. Laparoscopic or open gastrojejunostomy has been proposed as the treatment of choice in patients with advanced unresectable distal stomach tumor presenting with symptoms of GOOS [3,4]. Noticeably, laparoscopic gastroenterostomy might be difficult to be performed in a hostile abdomen because of the involvement of the root of the mesentery, infiltration of the surrounding structures, and peritoneal carcinosis [5]. Furthermore, laparoscopic or open gastrojejunostomy provides suboptimal palliation, because it is associated with postoperative complications ranging from 15% to 50% related to a delayed gastric emptying and a protract postoperative hospital stay. These results negatively affect the quality of life (QoL), and therefore, the efficacy of gastroenterostomy for palliation has been questioned [2,6,7]. In 1997, Kaminishi et al. [8] introduced a technique of stomach-partitioning gastrojejunostomy (SPGJ), which divides the lower part of the stomach and connects the jejunum to the proximal part of the stomach while maintaining a tunnel that is 2 to 3 cm in diameter along the lesser curvature. [7,8] This technique theoretically provides some benefits: endoscopic evaluation of the tumor response to palliative chemotherapy and the possibility of repeated endoscopic local treatment on the tumor, prevention of ingested food retention in the distal part of the stomach, thus facilitating gastric emptying and improving patient’s QoL [8]. A current alternative to laparoscopic or open surgical approach to an advanced gastric tumor is the positioning of a self-expandable metal stent (SEMS) which offers many potential advantages: the avoidance of general anesthesia for a laparoscopic or open approach, a shorter hospital-stay and a minor patient postoperative discomfort [1,6,9]. The preliminary results of a prospective randomized longitudinal cohort trial, comparing the QoL of patients affected with stage IV antropyloric stomach cancer and symptoms of GOOS who underwent endoscopic placement of SEMS or open SPGJ are herein presented.

## 2. Materials and Methods

In this preliminary study we examined the impact of SEMS positioning and open SPGJ on QoL in a cohort of 27 patients affected with stage IV antropyloric gastric cancer enrolled in a prospective randomized longitudinal cohort trial (Trial identifier NCT04599179). Study design details are available online (www.clinicaltrials.gov). Patients were randomly assigned into two treatment groups: Group 1 (13 patients) who underwent placement of SEMS and Group 2 (14 patients) in whom open SPGJ was performed. The random numbers were computer-generated. Data from patients followed the principles laid down in the Declaration of Helsinki and a formal ethic approval from our Institutional Review Board of Department of Surgery “Pietro Valdoni”, Sapienza University of Rome was obtained. A written informed consent for the treatment and the analysis of data for scientific purpose was obtained from all patients. The assistance of a psychologist was required for all patients because the patients were acknowledged of their terminal disease. We created a computerized database to collect clinical, pathological, intra- and post-operative outcomes, Karnofsky performance status scale [10], Visual Analogue Scale [11], quality of life (EQ-5D™) [12] and long-term survival. Age less than 85 years, pre-treatment histological diagnosis of gastric adenocarcinoma, computed tomographic (CT), palliative chemotherapy regimen, symptoms of GOOS (symptoms of GOOS include: regular, frequent feeling of bloating or fullness; feeling full after eating less food; nausea and vomiting of undigested food, especially right after eating, abdominal pain) lumen reduction ranging between 70% and 99% at gastroscopy were the inclusion criteria. A white blood cells count less than 4000/L, a platelet counts less than 70,000/L, patients with renal failure (i.e., albumin to creatinine ratio > 30 mg/mmol and estimated glomerular filtration rate <30–44 mL/min/1.73 m^2^), patients with major alterations of liver function tests (i.e., total bilirubin > 25.5 μmol/L, AST > 50 U/L, ALT > 66 U/L, PT-INR > 1.5). were the exclusion criteria. In the period of the present trial, 36 patients presented with Stage IV gastric cancer and symptoms of GOOS but 27 were enrolled. Five patients were excluded because they underwent conventional laparoscopic or open gastrojejunostomy or feeding jejunostomy. 

### 2.1. Endoscopic Procedure

The procedure was performed with the patient kept under light sedation with benzodiazepine administered according to his/her body weight. A subtotal tumor obstruction was always identified, and its length was measured using an adult or pediatric (4.8 mm in maximum diameter) endoscope (GIF-160 or GIF-XT Q160 Olympus; Olympus Italia, 20090, Segrate (MI), Italy). Conversely, a water-soluble contrast was injected under fluoroscopic control to define its length in the case of not superable stenosis. A biliary guidewire (Jagwire, Boston Scientific), passing in the working channel of the endoscope, was always progressed cautiously through the obstruction down to the duodenum. Its passage was followed under fluoroscopy. A covered (Ultraflex Covered Stent System, Microvasive; Boston Scientific Corporation, Boston, MA, USA) or uncovered (Wallstent Boston Scientific Corporation, Boston, MA, USA) SEMS, 10 cm in length was then slowly expanded, a 20 mm tumor-free margin above or below the tumor for a better anchoring was always obtained. The correct position was checked under fluoroscopy. A plain abdominal x-ray was performed the day after the procedure to exclude perforation or migration.

### 2.2. Radio-Chemotherapy

All clinical cases were discussed in a multidisciplinary team meeting. Based on previous therapy and according to the Eastern Cooperative Oncology Group (ECOG) performance status [13] and medical comorbidities, treatment included palliative chemo-radiotherapy or palliative systemic therapy alone. A central venous access was placed for the administration of chemotherapy. If not previously received, concomitant 5-fluoruracil (5-FU) with external beam radiotherapy was evaluated in those patients with performance status ≤1. First-line chemotherapy regimen was 5-FU with oxaliplatin (OXP). A 3-drug cytotoxic regimen, including 5-FU, OXP and taxane, was proposed to fit patients with performance status ≤1. Trastuzumab was added in case of HER2 overexpressing adenocarcinoma. All patients were treated until further disease progression or occurrence of unacceptable toxicity. In case of painful bone metastasis, palliative radiotherapy was performed to achieve pain relief.

### 2.3. Karnofsky Performance Scale and Quality of Life (QoL) Assessment

The patients’ functional status and QoL were evaluated before treatment and at 1, 3, and 6 months. The patients’ functional status was studied with the Karnofsky performance scale [10]. The QoL was studied with the EQ-5D-5L™ questionnaire (©EuroQol Group, Rotterdam, The Netherlands) [12] involving the following areas of investigations (mobility, self-care activities, usual activities, pain/discomfort, and anxiety/depression) with each dimension graded into five levels (i.e., no problem/slight problem/moderate problem/severe problem/extreme problem). The resulting health state was defined by a five-digit number combining one level from each of the five dimensions and then converted into the EQ-5D-5L index values according to the data from general age matched Italian population [12]. Furthermore, to help patients say how good or bad their health state was, a visual analog scale (EQ-VAS™) [11] with endpoints labelled “the best health you can imagine” graded 100 and “the worst health you can imagine” graded 0 was further analyzed.

### 2.4. Follow-Up Evaluation

Patients were followed-up for the entire length of the study on outpatient basis at 1, 3, and 6 months with hematochemical tests, chest X-ray, and abdominal CT scan.

### 2.5. Statistical Analysis

Our data were analyzed with a computer software program (SPSS Ver. 25.0.0.2; SPSS Chicago, IL, USA for MacOS High Sierra ver. 10.13.4, Apple Inc. 1983-2018 Cupertino, CA, USA). Due to sample sizes, non-parametric tests were applied. The Mann-Whitney U test analyzed the continuous variables whereas the Chi-square test with the 95% confidence interval reported for the size of the effect, i.e., the difference between mean values (C.I.) or the Fisher’s exact test explored the categorical variables. Due to the non-normally distribution of the sample, data were expressed as median, interquartile range (IQR), and mode [14]. Actuarial survival rate was assessed by the Kaplan-Meier method at 1-year. Standard error (SE) of survival rate were estimated at each censored case. Actuarial survival was limited at 1-year because analysis of a longer time period was statistically inappropriate for the small number of patients and the consequent high standard deviations. Differences with α-level of <0.05 were considered statistically significant.

## 3. Results

### 3.1. Demographics and Clinical Findings

Seventeen (63%) patients were males and 10 (37%) females with a median age at presentation of 76 years (min. 55 – max. 89; mode 60; IQR 8). Table 1 describes the demographic and clinical data of our series. Marital, social status, and level of education were similar among the groups. Specifically, 20 (74%) patients were married, 5 (18%) divorced, and 2 (8%) widowed. No patients belonged to the upper class whereas 6 (22%) to the upper-middle, 10 (37%) to the middle, 9 (33%) to the working and 2 (8%) to the lower classes. The majority of our patients (18–67%) had a basic education (primary, intermediate, and secondary) whereas 9 (33%) had college or master education.

### 3.2. Early Results

There was no postoperative mortality, but 1 (7%) major complication (pulmonary embolism successfully treated with anticoagulation and vena removable cava filter positioning) within 30 days in Group 2 was registered. Three (21%) patients of Group 2 had 1 superficial wound infection, 1 delayed oral feeding, and 1 bleeding from the superficial wound which spontaneously resolved. Patients belonging to Group 1 had no major or minor complications. The overall incidence of complications in Group 2 was significantly higher when compared to Group 1 (*p* = 0.037). Median length of stay was 8 days (min. 2 to max. 12; mode 4; IQR 6), but Group 2 patients had a significantly longer length of stay (*p* = 0.001; CI: 0.61 to 4.36) as compared to Group 1.

### 3.3. Long-Term Results

No patients were lost to a median follow-up of 9 months (min. 4 to max. 18; mode 8, IQR 3). There were no major or life-threatening complications related to chemotherapy but 5 (18%) patients (2 patients in Group 1 and 3 in Group 2) stopped chemotherapy because a significant deterioration of the liver function tests after the first or second cycle. Symptoms, potentially related to chemotherapy (fatigue, partial hair loss, decreasing liver function) were common (21–77% patients), and equally distributed in the two groups (12 in Group 1 and 9 Group 2). At a median follow-up of 4 months (min. 3 to max. 7; mode 3; IQR 2), 6 (46%) patients of Group 1 had SEMS re-dilatation for extrinsic stent tumor compression whereas none of Group 2 had anastomotic tumor compression or invasion (*p* = 0.004). SEMS re-dilatation for extrinsic tumor compression was always performed in outpatient basis; no complications were recorded. One-year actuarial survival rate of Group 1 was 34% (SE = 0.26; CI = 8.98 to 15.02) whereas in Group 2 was 38% (SE = 0.29; CI = 12.50 to 15.50).

### 3.4. Karnofsky Performance Scale and Quality of Life (QoL)

Karnofsky performance scale classifying patients’ functional impairment is shown in Figure 1, Panel A. No preoperative significant differences among the two groups (*p* = 0.749; CI: −9.42 to 12.94) were observed. At 1-month, Group 1 had a better performance scale with respect to Group 2 (*p* = 0.039; CI: 0.58 to 20.52), no differences were recorded at 3-month (*p* = 0.222; CI: −3.33 to 13.66), and at 6-month a significant improvement in Group 2 when compared to Group 1 (*p* = 0.012; CI: 7.71 to 12.51) was observed. Figure 1, Panel B shows the index values, harmonized to the Italian population. Preoperative index values were similar among the groups (*p* = 0.721; CI: −0.081 to 0.057), at 1-month, a statistically significant deterioration of the QoL in Group 2 when compared to Group 1 was observed (*p* = 0.004; CI: 0.044 to 0.21) whereas at 3-month no statistical differences among the groups were recorded (*p* = 0.19; CI: −0.026 to 0.12). Noticeably, at 6-month we observed a significant deterioration of the index values in Group 1 when compared to Group 2 (*p* = 0.009; CI: 0.25 to 0.43). Figure 1, Panel C shows the visual analog scale. Preoperatively no differences among the groups were recorded (*p* = 0.363; CI = −7.83 to 2.99), at 1-month Group 1 had a significant deterioration of QoL when compared to Group 2 (*p* = 0.035; CI = 0.45 to 11.10). At 3-month no differences were recorded (*p* = 0.314; CI = −2.39 to 7.15), and at 6-month Group 2 had a better visual analog scale when compared to Group 1 (*p* = 0.036; CI = 9.90 to 11.36).

## 4. Discussion

The influence of palliation in terms of QoL for the treatment of occlusive symptoms in the presence of advanced antro-pyloric tumors is under debate and marginally investigated [1,2,6]. In the present preliminary randomized trial, we answered several important unsolved questions regarding the best treatment to offer to patients affected with an advanced antro-pyloric tumor. Despite the increasing public attention to screening and significant awareness of the importance of an early diagnosis, more than 20% of patients with gastric cancer have at presentation an advanced disease [1]. Traditionally, laparoscopic or open gastrojejunostomy has been proposed as the treatment of choice in patients with advanced unresectable antro-pyloric tumors presenting with symptoms of GOOS [6,9]. Kumagai K. et al. [7], in a recent meta-analysis comparing partial stomach partitioning gastrojejunostomy versus conventional gastrojejunostomy for malignant gastroduodenal obstruction demonstrated that partial stomach partitioning offered several advantages in terms of preventing delayed gastric emptying, improving postoperative oral intake, and recovery compared to conventional gastrojejunostomy. In our preliminary study, we found that SEMS positioning avoids patient discomfort related to general anesthesia, nasogastric tube and laparotomy, parenteral feeding and bed stay resulting in an early gastrointestinal function recovery. Furthermore, a minimally invasive palliative treatment might be perceived, at least theoretically, by the patients more satisfying than a surgical operation. On the other hand, a gastrojejunostomy, even if it is a palliative operation, might give the feeling to have done something more with respect to SEMS positioning. Consequently, SEMS permits a rapid return to oral feeding as also reported in the recent literature [2,6,9,15,16,17,18,19,20,21,22,23,24,25,26,27,28] and confirmed by our investigation, thus accelerating the discharge of the patients from hospital and improving their return to a normal life. Furthermore, Ly J. et al. [29], in a systematic review, demonstrated a higher incidence of major medical complications (respiratory tract infection, myocardial infarction, and acute renal failure) in patients who underwent surgery for palliation of malignant GOOS when compared to SEMS positioning. Interestingly, these clear advantages of SEMS positioning are reflected in the better QoL at 1-month of these patients as compared to those who underwent surgical diversion. Conversely, at 6-months patients submitted to a surgical resection of the primary tumor, which improved their QoL with respect to SEMS [30].

Although this is not the aim of the present study, we tried to understand the underlying mechanisms affecting the reverse feeling of QoL in patients who underwent SEMS positioning at 6-month. We believe that the recurrence of the obstructive symptoms caused by stent obstruction from extrinsic tumor compression requiring endoscopic re-dilatation or stent repositioning might have affected the QoL. Conversely, an open SPGJ determined a better feeling of gastric emptying at the same interval follow-up because this approach is less subject to tumor compression or invasion. Noticeably, SPGJ can be also performed by a laparoscopic approach but we choose an open approach, because of the local invasion with infiltration of the surrounding structures, thus increasing the surgical risk of a laparoscopic approach. Obviously, a laparoscopic approach would have reduced the length of stay, the feeding resumption, the postoperative pain, the return to daily activities, thus most likely improving the QoL at 1-month, theoretically reaching the results of SEMS [3,4].

Our results are important for proposing an optimal therapeutic strategy. We believe that patients with acceptable general conditions, eligible for systemic chemotherapy should be submitted to SEMS positioning to avoid delay in chemotherapy administration, which at least theoretically can improve QoL and the survival interval; whereas surgery should be planned for patients with contraindication to chemotherapy in order to guarantee a better QoL.

This study has several limitations. Firstly, this is a preliminary study with a small number of patients. Secondly, life expectancy in patients with stage IV gastric cancer does not permit to perform a longer follow-up required to understand the factors influencing the QoL in accordance with the therapeutic choice. Third, our study was not blinded, and the patients were aware of the received treatment, and therefore those having SEMS insertion might have been more satisfied by this minimally invasive procedure. Finally, our study did not investigate the relapse of obstruction symptoms and/or the alteration in food intake during the follow-up. Our multivariate analysis confirmed that the small number of patients did not permit to understand the role of the different variables inserted in the statistical model.

## 5. Conclusions

Patients affected with stage IV antro-pyloric cancer and symptoms of GOOS have an immediate postoperative, and at 30-day benefit in QoL after SEMS positioning but after 6-month the QoL deteriorates in this group of patients. Conversely, a significant improvement in the patients submitted to SPGJ is recorded at 6-month. We do not have clear evidence able to justify this bimodal fluctuation, but we believe that a higher risk of hospital re-admission in the SEMS group might jeopardize the QoL at a mid-term follow-up.

## Figures and Tables

**Figure 1 medicina-57-00428-f001:**
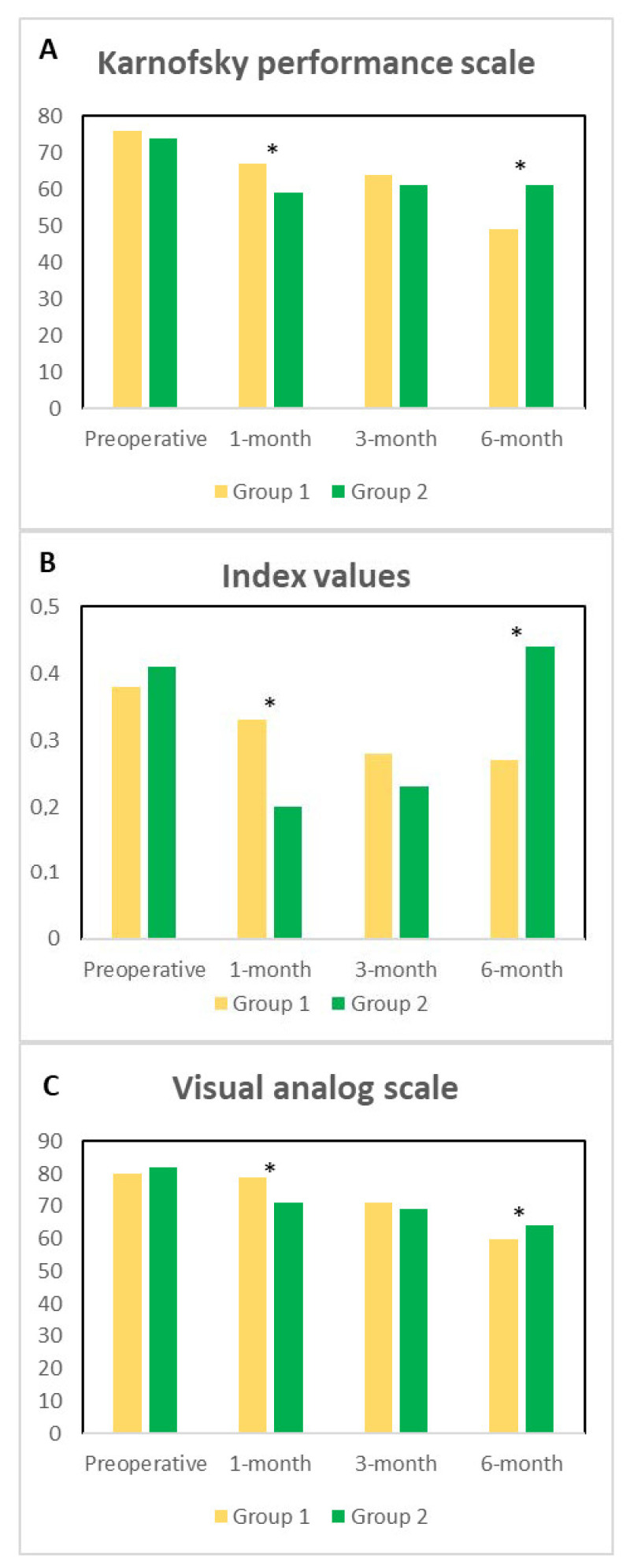
Panel (**A**). Preoperative, at 1-, 3- and 6-month Karnofsky performance scale among the groups of patients is described. Panel (**B**). Preoperative, at 1-, 3- and 6-month Index values among the groups of patients is described. Panel (**C**). Preoperative, at 1-, 3- and 6-month Visual analog scale among the groups of patients is described. The significances among the groups are depicted with asterisks.

**Table 1 medicina-57-00428-t001:** Demographics and clinical data.

	Group 1	Group 2	Significance
Number	13	14	
Median age (Mode; IQR)	77 (80; 8)	73 (81; 10)	0.201
Sex (M/F)	8/5	9/5	0.598
Total bilirubin (μmol/L)	13 (4)	11 (3)	0.439
AST (U/L)	27 (7)	31 (9)	0.829
ALT (U/L)	45 (9)	41 (10)	0.377
PT-INR	0.9 (0.8)	1.1 (0.7)	0.568
Ascites	2	2	0.875
Liver metastasis	11	13	0.773
Pulmonary metastases	4	3	0.263

IQR = interquartile range; AST = aspartate AST = aspartate aminotransferase; ALT = alanine aminotransferase; PT-INR = prothrombin time-international normalized ratio.

## Data Availability

https://www.clinicaltrials.gov/ct2/results?cond=&term=NCT04599179&cntry=&state=&city=&dist=.

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
