# Peer review of "Palliative Surgery or Metallic Stent Positioning for Advanced Gastric Cancer: Differences in QOL"

_medicina, 2021, doi:10.3390/medicina57050428_

Round 1
Reviewer 1 Report
This study has several weak points that need attention. The methodology should be more precisely presented. The nature of the study, inclusion and exclusion criteria, type of randomization, blindness of treatment received, variables studied at inclusion and at follow-up, proper statistics and analysis should be clear and accurately given. The results need to be presented in a more clear and informative way. The discussion should be more extensive and comprehensive and discuss critically the results along with possible factors affecting the findings. The weak points, possible bias and the possible reasons for these findings should be also presented. Same more specific comments include:
1) Line 80. … adjuvant-neoadjuvant chemotherapy regimen ….
adjuvant-neoadjuvant to what? There was no surgical; treatment in these patients.
2) Line 91. open gastroenterostomy or digiunostomy.
What do you mean by digiunostomy? In addition, it is better to be consistent and use the same term throughout the manuscript i.e. either gastrojejunostomy or gastroenterostomy.
3) Lines 92-95. You do not state that all patients with stage IV disease and concomitant GOOS were evaluated. Therefore, it is not necessary to report patients who did not met the prospective and preset criteria for inclusion in the study.
4) Materials and Methods. Line 68, Patients were randomly assigned into two treatment groups and Lines 95-96, patients were randomly assigned to one of the two treatment groups.
The method of randomization is not given and the objectivity of selection and assignment to each group is not evident. This is essential for a randomized prospective study.
5) Line 104. A 20 mm tumor-free margins above and below were … (Please check grammar) and Line 109, the statement that a 10 cm SEMS in length was inserted in all patients.
What was the point to measure the 20mm tumor-free margin on both sides since the same length of SEMS was used in all patients?
6) Line 141, Patients were followed-up on outpatient basis at 3 and 6 months
How did you then evaluate the Karnofsky Index and the QoL at 1 month?
7) Line 149. Due to the heterogeneity of the sample, data were expressed as mean ± standard deviation,
Normally distributed data are presented as mean ± SD or SE, whereas non-normally distributed data are displayed as median (IQR or Range).
8) Results. Line 176. The overall incidence of complications in Group 2 was significantly higher when compared to Group 1 (complications in Group 1 are not shown), Line 179, significantly higher length of stay (longer length of stay), Line 184, but 5 (18%) patients (3 patients in Group 1 and 3 in Group 2) stopped (3+3, that makes 6 patients), Line 187, and equally distributes in the two groups (distributed).
Overall, the Results section needs to be more succinct and clearly presented. The studied variables and the corresponding figures and statistics should be given in a Table.
9) Line 245. surgery for palliate malignant GOOS (palliative or for palliation of).
10) Conclusions. Line 277, The QoL of patients is significantly better in patients treated with SEMS positioning but (this not entirely true).
Overall the conclusions section should be rephrased in a way reflecting accurately the actual findings of the study and the conclusion based on them.
11) How confident are the authors for their results especially to the observed differences at the 1-month time point. This not a blind study, patients were aware of the received treatment and therefore those having SEMS insertion might have been more satisfied by the minimally invasive procedure. This should be commented in the discussion. In addition, the need for endoscopic re-intervention and the reasons for that during the follow-up is not clear and unfortunately factors related to the efficacy of each technique and affecting the results of the study such as relapse of obstruction symptoms and/or alterations in food intake were not studied during the follow-up. This information is important for the explanation of the results and these points should be discussed in the discussion section of the manuscript.
Author Response
REVIEWER 1
We are very pleased to revise the manuscript according to your constructive criticisms.
This study has several weak points that need attention. The methodology should be more precisely presented. The nature of the study, inclusion and exclusion criteria, type of randomization, blindness of treatment received, variables studied at inclusion and at follow-up, proper statistics and analysis should be clear and accurately given. The results need to be presented in a more clear and informative way. The discussion should be more extensive and comprehensive and discuss critically the results along with possible factors affecting the findings. The weak points, possible bias and the possible reasons for these findings should be also presented. Same more specific comments include:
As you suggested the entire manuscript was revised according to your suggestions.
1-Line 80. … adjuvant-neoadjuvant chemotherapy regimen …. adjuvant-neoadjuvant to what? There was no surgical treatment in these patients.
We agree with your comment and to better specify this issue the manuscript was modified as it follows: “palliative chemotherapy regimen”. (Line 81)
2-Line 91. open gastroenterostomy or digiunostomy. What do you mean by digiunostomy? In addition, it is better to be consistent and use the same term throughout the manuscript i.e. either gastrojejunostomy or gastroenterostomy.
We agree with your comment and to better specify this issue the manuscript was modified (Line 92) and the other changes are highlighted in red throughout the text.
3-Lines 92-95. You do not state that all patients with stage IV disease and concomitant GOOS were evaluated. Therefore, it is not necessary to report patients who did not met the prospective and preset criteria for inclusion in the study.
As you suggested lines 92-95 were deleted.
4-Materials and Methods. Line 68, Patients were randomly assigned into two treatment groups and Lines 95-96, patients were randomly assigned to one of the two treatment groups. The method of randomization is not given and the objectivity of selection and assignment to each group is not evident. This is essential for a randomized prospective study.
As you suggested the text was modified as it follows: “The random numbers were computer-generated.”.
5-Line 104. A 20 mm tumor-free margins above and below were … (Please check grammar) and Line 109, the statement that a 10 cm SEMS in length was inserted in all patients. What was the point to measure the 20mm tumor-free margin on both sides since the same length of SEMS was used in all patients?
As you suggested the text was modified as it follows: “… a 20 mm tumor-free margin above or below the tumor for a better anchoring was always obtained”. (lines 104-105)
6-Line 141, Patients were followed-up on outpatient basis at 3 and 6 months. How did you then evaluate the Karnofsky Index and the QoL at 1 month?
We corrected this issue in the text. (Line 136)
7-Line 149. Due to the heterogeneity of the sample, data were expressed as mean ± standard deviation,
Normally distributed data are presented as mean ± SD or SE, whereas non-normally distributed data are displayed as median (IQR or Range).
We agree with your comment and data throughout the text are now expressed as median, IQR, and mode.
8-Results. Line 176. The overall incidence of complications in Group 2 was significantly higher when compared to Group 1 (complications in Group 1 are not shown), Line 179, significantly higher length of stay (longer length of stay), Line 184, but 5 (18%) patients (3 patients in Group 1 and 3 in Group 2) stopped (3+3, that makes 6 patients), Line 187, and equally distributes in the two groups (distributed). Overall, the Results section needs to be more succinct and clearly presented. The studied variables and the corresponding figures and statistics should be given in a Table.
As you suggested the text was modified accordingly (Line 176, Line 179, Line 184, and Line 187). To better elucidate our results, section 3.4 was completely rewritten, and Figure 1 Panel A-C was added.
9-Line 245. surgery for palliate malignant GOOS (palliative or for palliation of).
As you suggested the text was modified accordingly.
10-Conclusions. Line 277, The QoL of patients is significantly better in patients treated with SEMS positioning but (this not entirely true). Overall the conclusions section should be rephrased in a way reflecting accurately the actual findings of the study and the conclusion based on them.
We agree with your comment and the section was completely revised (Lines 274-280).
11-How confident are the authors for their results especially to the observed differences at the 1-month time point. This not a blind study, patients were aware of the received treatment and therefore those having SEMS insertion might have been more satisfied by the minimally invasive procedure. This should be commented in the discussion. In addition, the need for endoscopic re-intervention and the reasons for that during the follow-up is not clear and unfortunately factors related to the efficacy of each technique and affecting the results of the study such as relapse of obstruction symptoms and/or alterations in food intake were not studied during the follow-up. This information is important for the explanation of the results and these points should be discussed in the discussion section of the manuscript.
We are confident of our results at 1 month, the use of SEMS or SPGJ might give different psychologic perception in the patient’s mind. This was not the aim of the present study but to better clarify the text was accordingly modified (Lines 229-233). As you suggested the discussion section was modified to better elucidate the limitations of our study (Lines 266-271).
Reviewer 2 Report
The subject of the article is extremely interesting and of actuality. The paper is well written, the methodology is clearly described and the conclusions are sustained by the results
Major points:
- The study groups are composed of a small number of patients (13 in Group 1 and 14 in Group 2). However, as the subject is relatively new, the experience is important for the public and could be included in future systematic reviews
- 12 out of the 22 references are articles of the main author. Even the citations are relevant and reflects a permanent preoccupation of the authors for the subject, the proportion of self-citation is too high. The discussions should be enlarged with references to other bibliographic data.
- A better comparison between the outcomes of the 2 methods would be useful for the reader in order to understand the benefits and limits of the SEMS as opposed to open GEA: days of hospitalization in group 2, secondary hospitalizations for the 6 patients in group 1 – with days of hospital stay, reasons for reintervention, complications after reintervention if any. Are those 6 patients those with lower scores of quality of life in longer-term? Are there other reasons identified for lower QoL in the patients in group 1 who did not undergo reintervention?
A table with comparative data may be useful for the readers
Minor point:
R 91: please correct…… digiunostomy
R 161: please correct… Maritial, social ..
R 272: please correct … The Our multivariate
Author Response
REVIEWER 2
We are very pleased to revise the manuscript according to your constructive criticisms.
The subject of the article is extremely interesting and of actuality. The paper is well written, the methodology is clearly described and the conclusions are sustained by the results
Major points:
1-The study groups are composed of a small number of patients (13 in Group 1 and 14 in Group 2). However, as the subject is relatively new, the experience is important for the public and could be included in future systematic reviews
We are grateful for your kind comment and we hope that our study will be included in a systematic review.
2-12 out of the 22 references are articles of the main author. Even the citations are relevant and reflects a permanent preoccupation of the authors for the subject, the proportion of self-citation is too high. The discussions should be enlarged with references to other bibliographic data.
We greatly enlarged the reference list with other bibliographic data.
3-A better comparison between the outcomes of the 2 methods would be useful for the reader in order to understand the benefits and limits of the SEMS as opposed to open GEA: days of hospitalization in group 2, secondary hospitalizations for the 6 patients in group 1 – with days of hospital stay, reasons for reintervention, complications after reintervention if any. Are those 6 patients those with lower scores of quality of life in longer-term? Are there other reasons identified for lower QoL in the patients in group 1 who did not undergo reintervention?
A table with comparative data may be useful for the readers
As also suggested more details on the 6 patients with SEMS obstruction for tumor growth compression are now added (Line 185-186). by reviewer #1 the results section para 3.4 was completely rewritten and Figure 1 was added.
Minor point:
1-R 91: please correct…… digiunostomy: Done
2-R 161: please correct… Maritial, social. Done
3-R 272: please correct … The Our multivariate: Done
Round 2
Reviewer 1 Report
The authors responded sufficiently to the comments raised by the reviewers and the manuscript has improved. I have no any further comments.
Reviewer 2 Report
All the comments have been addressed. The article may be published